# YB1 Is a Major Contributor to Health Disparities in Triple Negative Breast Cancer

**DOI:** 10.3390/cancers13246262

**Published:** 2021-12-14

**Authors:** Priyanka Shailendra Rana, Wei Wang, Akram Alkrekshi, Vesna Markovic, Amer Khiyami, Ricky Chan, Adam Perzynski, Natalie Joseph, Khalid Sossey-Alaoui

**Affiliations:** 1Department of Medicine, MetroHealth Medical Center, Cleveland, OH 44109, USA; pxr240@case.edu (P.S.R.); wxw363@case.edu (W.W.); axa1061@case.edu (A.A.); adam.perzynski@case.edu (A.P.); njoseph@metrohealth.org (N.J.); 2Case Western Reserve University School of Medicine, Case Western Reserve University, Cleveland, OH 44016, USA; vxm196@case.edu (V.M.); axk11@case.edu (A.K.); 3Case Comprehensive Cancer Center, Case Western Reserve University, Cleveland, OH 44106, USA; erc6@case.edu

**Keywords:** YB1, WAVE3, TNBC, chemoresistance, cancer stem cells, cancer disparities

## Abstract

**Simple Summary:**

Triple negative breast cancer (TNBC) is a devastating disease that affects many women, due to the lack of FDA-approved targeted therapy. In the absence of cell surface receptors ER, PR, and Her2 that can be targeted with hormonal and antibody treatments, cytotoxic chemotherapy remains the major course of treatment, with a dismal response and rapid recurrence due to the acquisition of resistance. TNBC is also twice as more prevalent in African American (AA) when compared to Caucasian American (CA) women. This study investigated the role of the YB1 gene in the disparities in TNBC between AA and CA women. We found that YB1 is highly expressed in TNBC tumors of AA origin when compared to CAs. Increased expression levels and activity of YB1 correlates with poor disease outcomes, resistance to chemotherapy, and the activation of the cancer stem cell (CSC) phenotype, with higher levels in AA than in CA TNBC tumors. More importantly, we found that the targeted inhibition of the expression and activity of YB1 significantly inhibited the oncogenic behavior of AA tumors through sensitization to chemotherapy and inhibition of CSCs. Our study is the first to show that YB1 activity may be a major biological contributor to the health disparities in TNBC, and that development of therapies that specifically target YB1 could reduce these disparities.

**Abstract:**

Triple negative breast cancer (TNBC) is the most aggressive amongst all breast cancer (BC) subtypes. While TNBC tumors represent less than 20% of all BC subtypes, they are responsible for the most BC-related deaths. More significantly, when considering TNBC incidence across all racial/ethnic groups, TNBC accounts for less than 20% of all BCs. However, in non-Hispanic black women, the incidence rate of TNBC is more than 40%, which may be a contributing factor to the higher BC-related death rate in this population. These disparities remain strong even after accounting for differences in socioeconomic status, healthcare access, and lifestyle factors. Increased evidence now points to biological mechanisms that are intrinsic to the tumor that contribute to disparate TNBC disease burdens. Here, we show that YB1, a multifunction gene, plays a major role in the TNBC disparities between African American (AA) and Caucasian American (CA) women. We show in three independent TNBC tumors cohorts, that YB1 is significantly highly expressed in AA TNBC tumors when compared to CAs, and that increased levels of YB1 correlate with poor survival of AA patients with TNBC. We used a combination of genetic manipulation of YB1 and chemotherapy treatment, both in vitro and in animal models of TNBC to show that YB1 oncogenic activity is more enhanced in TNBC cell lines of AA origin, by increasing their tumorigenic and aggressive behaviors, trough the activation of cancer stem cell phenotype and resistance to chemotherapeutic treatments.

## 1. Introduction

In women, ~90% of breast cancer-related deaths are caused by metastasis [1]. In 2021, breast cancer (BC) surpassed all other cancers affecting women, and ranks number two as the cause of cancer-related deaths in women [2]. BC is a heterogeneous disease that can be divided into at least five genetically distinct subtypes, where more than 10 subtypes can be identified based on their molecular signatures [2,3,4,5,6,7]. Triple-negative BC (TNBC), a BC subtype that lacks the expression of hormone receptors (ER-a and PR) and ErbB2/HER2 [8,9,10,11,12], is notoriously known for its aggressiveness and lethality because of its high metastatic and recurrence potentials, coupled with the lack of FDA-approved targeted therapies that could be effective against this BC subtype. Additionally, when TNBC tumors recur, they are generally associated with the acquisition of resistance to standard-of-care chemotherapies through mechanisms that remain to be resolved. Interestingly, while across all racial/ethnic groups, TNBC accounts for less than 20%, its incidence rate more than doubles in non-Hispanic black women, which may contribute to the higher breast cancer death rate in this population. There is now increasing evidence that African American (AA) women with TNBC have worse clinical outcomes when compared with Caucasian American (CA) women who have the same disease. At MetroHealth Medical Center, the site of this study and the largest SafetyNet hospital in north-east Ohio that is also serving the most disadvantaged populations in the Cleveland area, 39% of our patients treated for BC are AAs when compared to the national average of 11%. In fact, of the 113 women with TNBC that were treated at MetroHealth Medical Center between 2014 and 2018, 58 (51%) were AAs. These statistics, that are characteristic of our sociodemographic area, place us in a unique position to study cancer health disparities in our TNBC patient population. More importantly, the disparities in TNBC still remain poorly understood, in part due to the complex interactions between genetics, social environment, and lifestyle that may contribute to the observed differences. Moreover, there is growing evidence that the observed disparities in TNBC outcomes between AAs and CAs are not exclusively the result of socioeconomic or healthcare access factors, but rather reflect biological mechanisms intrinsic to the tumor that readily contribute to unequal TNBC disease burdens [13,14,15]. 

The Y Box binding protein 1 (YB1) is a multifunctional protein, found both in the cytoplasm and inside the nucleus, that belongs to the highly conserved Cold Shock Domain protein family [16]. Among its multitude of functions, YB1 is more known to be involved in the regulation of transcription, mRNA stability, and splicing [17,18,19,20,21,22,23,24,25,26,27,28]. More recently, YB1 was found to play key roles in the regulation of tumor development, progression and metastasis, as well as drug resistance in several cancers, including the ones generated in the breast [17,24,26,29,30,31,32,33,34,35,36,37,38,39,40,41,42,43,44,45]. In this study, we show that dysregulated YB1 expression to be associated primarily with breast cancer cell lines in primary tumors of the TNBC subtype as compared to their luminal or HER2+ counterparts. We also show that YB1 abundance correlates with the metastatic potential of TNBC cell lines, and robust YB1 expression correlates with reduced overall survival in BC patients. We also show that YB1 expression levels to be significantly (*p* < 0.01) higher in TNBC cell lines and tumors of AA origin when compared to their CA counterparts. Importantly, this association remained significant even after controlling for known prognostic factors, such as tumor stage, age, and node status, suggesting that YB1 activation is likely to be a population-specific prognostic factor in TNBC. Mechanistically, we show that aggressiveness of TNBC tumors of AA origin is driven by the YB1-mediated activation of resistance to chemotherapeutic drugs, by activating genes that are associated with the cancer stem cell phenotype and chemoresistance. Our data identify YB1 as a major biological factor of cancer health disparities in AA women with TNBC tumors, and open opportunities for the development of targeted therapies against TNBC tumors that are disparately affecting AA women.

## 2. Materials and Methods

### 2.1. Ethics Statement

All animal studies were performed under protocols approved by the Institutional Animal Care and Use Committee and conducted in accordance with the guidelines and regulations set and approved by the MetroHealth Medical Center, Case Western Reserve University, and NIH. For this study, we used six to eight week-old female NSG or Balb/C mice (Jackson Laboratory, Farmington, CT, USA).

### 2.2. Cell Lines and Reagents

TNBC cell lines 4T1, MDA-MB-231, MDA-MB-468, HCC1806, HCC38, HCC70, and BT549 were obtained from American Type Culture Collection (ATCC) and maintained according to the manufacturer’s protocols. Cell lines were also routinely authenticated by STR DNA fingerprinting analysis. The Yb1-KO cells were generated by electroporation as described below.

### 2.3. Antibodies and Reagents

We used the following antibodies: rabbit antibodies against pAKT-S473 (1:1000), pAKT-T308 (1:1000), AKT (1:1000), p-Yb1 (Cell Signaling Technology, Danvers, MA, USA) (1:1000); Yb-1 (Abcam, Cambridge UK) (1:1000); goat horseradish-peroxidase-conjugated anti-mouse IgG and goat horseradish-peroxidase-conjugated anti-rabbit IgG (Biorad, Hercules, CA, USA) (1:2000) and mouse monoclonal anti-actin (Sigma, St. Louis, MO, USA) (1:5000). ECL reagent was from Thermo Scientific (Hanover Park, IL, USA). The antibodies were dissolved to a working concentration either in 5% BSA (Primary antibodies) or 5% non-fat dry milk (secondary antibodies). Cisplatin was obtained from Sigma and doxorubicin was obtained from European Pharmacopoeia, and used at 30 μM and 200 nM, respectively, to treat cell cultures for 48 h.

### 2.4. Immunoblotting

Immunoblotting analyses were performed as described previously [46,47]. Cells cultured on 6-well plates were washed with ice-cold PBS twice before lysis. Ice-cold RIPA lysis buffer with protease and phosphatase inhibitors was used to lyse the adherent cells and the lysate was transferred to prechilled 1.5 mL microcentrifuge tubes. The samples were then centrifuged at 15,000 RPM for 20 min at 4 °C. Cell lysates with an equal amount of total protein (25 mg) were first digested in SDS sample buffer, then resolved on a 10% SDS-polyacrylamide gel, followed by transferring the proteins to PVDF membrane. The membrane was incubated in a blocking solution (5% BSA) to prevent non-specific binding at room temperature for 60 min. After blocking, the membrane was then incubated with primary antibody overnight at 4 °C followed by washing and incubating again for an hour with an appropriate secondary antibody at room temperature. The membrane was washed, and signals were developed with a Pierce ECL Western Blot chemiluminescence detection kit and further visualized and imaged using ChemiDoc MP Imaging System. Signals were quantified using ImageJ software, according to the parameters described in ImageJ user guide (http://rsbweb.nih.gov/ij/docs/guige/146.html, accessed on 2 July 2021. Average values from 3 different blots are presented.

### 2.5. RNA Extraction and Quantitative Real Time Reverse Transcription PCR (qt-RT-PCR)

RNA was extracted from cell lines as described previously, according to the manufacturer’s instructions [48]. Cells were lysed in a 6 well plate using TRIzol reagent (Invitrogen), and total RNA resuspended in RNase-free water was quantified using a Nanodrop 2000 spectrophotometer (ThermoFisher Scientific, Waltham, MA, USA). To generate cDNA for qtRT -PCR, 1 mg RNA was used with the SuperScript III First-Strand Synthesis System RT-PCR kit (Invitrogen, Waltham, MA, USA). qtRT -PCR was done as described previously, using the C1000 Touch Thermal Cycler CFX96 Real-Time System (Bio-Rad, Hercules, CA, USA). Oligonucleotide primers used for qtRT-PCR were described in [49].

### 2.6. Colony Formation Assay

Clonogenic growth was assessed by seeding 5000 cells (for MDA-MB-231, MDA-MB-157, HCC70, HCC38) or 10,000 cells (for MDA-MB-468 and BT549) onto a 6 well plate and incubating them for 5 days until macroscopically visible colonies were formed. Growth media were replaced once every other day. Colonies were visualized by fixing cells in 4% PFA for 20 min, were washed with PBS, and then stained with 0.05% crystal violet [50]. Plates were washed with water, and colonies were counted under a dissecting scope.

### 2.7. Generation of Cisplatin- and Doxorubicin-Resistant TNBC Cells

MDA-MB-231 or MDA-MB-468 cells were cultured with increasing concentrations of either cisplatin or doxorubicin over a 60-day period, starting with 1 µM for 7 days, 10 µM for 14 days, 21 µM, 30 µM (MDA-MB-231), or 40 µM (MDA-MB-468) for 30 days, for cisplatin. For doxorubicin, cells were cultured with 50 nM for 7 days, 100 nM for 14 days, 150 nM for 21 days, followed by 300 nM (MDA-MB-231) or 450 nM (MDA-MB-468) for 30 days. After each cycle, the IC50 was determined by MTT to determine the concentration for the next cycle. After the 60 day period, resistance to cisplatin or doxorubicin was established and the derived cells were used for subsequent analyses. 

### 2.8. sgRNA Preparation and Electroporation for Generating eYb1KO Cell Lines

The sgRNA pool of 3 guide RNAs for mouse Yb1 (A*C*G*GGCAGCGGCGCGGGUAG, G*G*C*GGGGACAAGAAGGUCAU, and G*G*C*CCGAGCCACGGACUACG) and 2 guide RNAs for human Yb1 (U*U*U*UCCAGCAACGAAGGUUU and U*U*C*AUCAACAGGUGAGCUGC), with their respective Synthego modified EZ scaffolds targeting Yb1, were obtained from Synthego. The sgRNA pellets were rehydrated in 1X TE buffer (provided by Synthego, Menlo Park, CA, USA) to make a stock of 100 µM. A working solution of 30 µM sgRNA was made (in nuclease-free water) fresh before electroporating the cells. For every reaction, the RNP (Ribonucleoprotein) complex was assembled by adding 3 µl of 30 µM sgRNA to 0.5 µl of 20 µM Cas9 (provided by Synthego) at a ratio of 9:1 in 3.5 µl resuspension buffer R, provided by Neon Transfection System; Invitrogen Basel, Switzerland, and was incubated for 10 min at room temperature.

### 2.9. Generation of Yb-1 Knockout Cell Lines Using Electroporation

Electroporation of all 7 TNBC cell lines (4T1, MDA-MB-231, MDA-MB-468, HCC1806, HCC38, HCC70 and BT549) was achieved by an implemented electroporation device system, according to manufacturer’s instructions (Neon Transfection System; Invitrogen, Basel, Switzerland). The Neon Transfection System 10 µL kit was used for the transfection of human and mouse TNBC cells. Cells were subcultured 48 h before electroporation and harvested at nearly 80% confluency. Cells at a density of 2 × 10^5^ were washed with PBS and resuspended in 5 µl Resuspension buffer R (Neon Transfection System; Invitrogen Basel, Switzerland). Within 15 min of resuspension, the cells were added to the tube containing RNP and the cell-RNP complex was electroporated with the Neon Transfection System. Per electroporation, 2 × 10^5^ cells were taken up in a 10 µL Neon tip using the Neon Transfection System pipette (Invitrogen). The electroporation was performed by applying 4 pulses at 1400 Volts for 10 ms to all the TNBC cell lines. The respective control cells were incubated with the resuspension buffer without the sgRNA and electroporated at the same settings. After electroporation, the cells were seeded in 6 well plate by adding 2.5 mL DMEM (Cytiva) or RPMI (ATCC) with 10% FBS without antibiotic supplements. The cells were cultured for 48–72 h and subsequently proceeded for further analysis. We used Western Blot to verify the loss of YB1 expression.

### 2.10. Animal Experiments

For MDA-MB-231 and MDA-MB-468 human TNBC cell lines, parental (control) or Yb1-KO cells (1 × 10^6^ cells/injection) were injected into mammary fat pads in both sites (*n* = 10 tumors) of NSG mice (*n* = 5 mice). For the 4T1 murine TNBC cell lines, 100 K cells per injection were used to inject syngeneic BalbC mice. Mice were anaesthetized using isofluorane prior to injecting tumor cells. Tumor growth was followed by twice weekly monitoring of tumor volume with digital Vernier calipers. At the start of the experiment, mice were euthanized, and tumors were excised, weighed, and snap-frozen in liquid N2 and stored in a −80 °C freezer for subsequent analyses. Lung metastasis nodules were counted, and the results were plotted as the average number of foci per lobe. For the animal treatment with cisplatin, 21 days post injection of parental or cis-resistant MDA-MB-468 cells, when the first tumors grew to ~100 mm^3^, mice were then treated with cisplatin (1 mg/kg, IP) 2 times a week for 4 weeks. When tumor volume had reached 1000 mm^3^, all animals were euthanized.

### 2.11. Human TNBC Specimens

A total of 45 TNBC tumors (25 AA and 20 CA, self-reported ethnicity) were identified from the MetroHealth breast cancer tumor biobank of FFPE tumors. All FFPE TNBC tumor blocks were confirmed for tumor content and TNBC diagnosis by a subspecialty breast pathologist. Total RNA was isolated from the tumor sections using the Qiagen AllPrep FFPE DNA/RNA kit in accordance with manufacturer’s protocols. After RNA extraction, RNA purity was assessed using the 260/280 ratio, as well as the RNA integrity number (RIN). Only those RNA samples with an RIN > 7 were retained for subsequent qt-RT-PCR as described above.

### 2.12. Analyses of RNA-Seq Data from the TCGA Datasets

Data generated from the illumina sequencing platform were qualitatively assessed and trimmed to remove the adapter sequences, with the help of TrimGalore! v0.4.2 (Babraham Bioinformatics, Cambrigde, UK), a wrapper script for FastQC, and cutadapt. Sequence reads that passed quality control were aligned with the human reference genome (GRCh38) using the STAR aligner v2.5.1 [51]. The sequence alignment in this case was guided with the help of GENCODE annotation for hg38. The aligned reads were then analyzed for differential expression using an RNASeq analysis package—Cufflinks v2.2.1 [52], which reports the fragments per kb of exon per million fragments mapped (FPKM) for each gene. Cuffdiff was used to generate the differential analysis report. A significance cutoff of *q*-value < 0.05 (after Benjamini-Hochberg correction for multiple testing) was used to identify differential genes, which were then subjected to gene set enrichment and pathway analysis using iPathwayGuide (AdvaitaBio, Ann Arbor, MI, USA) to further determine any relevant biological processes that may be differentially represented between the control and the experimental groups. 

### 2.13. Bioinformatic Methods for Datamining of TCGA PanCancer Dataset

BC patient data from TCGA PanCancer Atlas [53] was accessed using cBioPortal [54,55]. The PanCancer Atlas includes 1082 patients with mRNA expression data and 151 individuals with data available for survival analysis. Race information for the samples was obtained through cBioPortal and 182 patients identified as African American and 749 identified as Caucasian. The distribution of BC subtypes by race is provided in Table 1. Gene expression differences between normal and tumor pairs were tested for significance by the Wilcoxon Signed Rank Sum test. Differences between subtypes were tested for significance using the Wilcoxon Rank Sum test. The mRNA co-expressions were reported and graphed for the genes of interest based on RSEM values which were batch-normalized from Illumina HiSeq RNASeq data. The protein expression levels were reported and graphed for the genes of interest based on protein expression (RPPA). Additionally, the Spearman correlation between genes of interest were reported, along with their *p*-value and multiple testing corrected q-value, obtained using cBioPortal. Overall survival graphs for thegenes of interest were generated using R (‘survival v3.2-3′ package from CRAN) using the survival data obtained from cBioPortal. 

### 2.14. Other Statistical Analyses

Experiments were done in triplicate and analyzed using the Student’s *t*-test. In calculating the two-tailed significance levels for equality of means, equal variances were assumed for the two populations. Results were considered significant at *p* < 0.05.

## 3. Results

### 3.1. YB1 Expression Levels Are Increased in TNBC Cell Lines and Tumors and Are Associated with Poor Survival and Worst Patients’ Outcomes

We previously reported that expression levels of YB1 were significantly elevated in aggressive BC cancer cell lines when compared to their less aggressive counterparts [31]. More importantly, we showed that YB1 expression levels are significantly higher in TNBC cell lines when compared to all other BC subtypes [31], which indicated an association between YB1 expression and the aggressiveness of TNBC cell lines. We also showed that YB1 expression correlated with the metastatic potential of three TNBC metastasis progression models, namely the human MCA10A series [56], the murine 4T1 series [57], and the NME series [58,59]. To validate these findings in human patients’ specimens, we interrogated the datasets of two large, independent BC cohorts. The first cohort, the TCGA PanCancer BC cohort, which contains clinical information on 1082 BC patients, including 181 TNBCs (Table 1), showed a significant (*p* < 0.0001) positive association between YB1 expression levels and the basal (TNBC) BC subtype (Figure 1A). Elevated expression levels of YB1 in the TNBC subtype also correlated (*p* = 0.03) with poor disease outcome and reduced survival probability in this cohort (Figure 1B). Next, we interrogated the BC KM-Plotter (https://kmplot.com/analysis, accessed on 6 January 2021) cohort, which contains clinical information on ~5000 BC patients, including 405 TNBCs, and this showed a very significant (*p* = 1.9 × 10^8^), positive correlation between YB1 mRNA expression and reduced survival probability (Figure 1C, upper panel). In this cohort, BC patients with high levels of YB1 in their tumors had worst clinical outcomes when compared to patients with low levels of YB1, and patients with high tumor YB1 have an average reduced survival of 31 months when compared to patients with low tumor YB1 (Figure 1C, lower panel). This correlation between YB1 expression levels and poor prognosis was also confirmed at the protein expression level (Figure 1D). Finally, we found YB1 mRNA expression levels in the TNBC cohort to also correlate with reduced survival (Figure 1E). Thus, the data support the function of YB1 as a promoter of BC aggressiveness in patients with TNBC tumors. 

### 3.2. YB1 Is Disparately Highly Expressed in AA vs. CA TNBC Cell Lines and Tumors and Correlates with Poor Outcome in AA TNBC

The literature provides overwhelming evidence that incidence of TNBC and TNBC-related death are disparately elevated in AA women when compared to their CA counterparts. The socioeconomic status and lack of access to health care by the AA population cannot alone explain these disparities, which suggests a difference in the biology of TNBC tumors between AA and CA might also contribute to the observed disparities. We therefore set out to investigate whether the oncogenic activity of YB1 in TNBC tumors contributes to these disparities. First, we assessed the expression levels of YB1 by RT-PCR in six established TNBC cell lines; three from AA and three from CA patients, and found YB1 to be significantly (*p* < 0.001) higher in AA cell lines MDA-MB-468, HCC70, and HCC180 when, compared to CA TNBC cell lines MDA-MB-231, BT549, and HCC38 (Figure 2A,B). We confirmed the results that were obtained in our cell lines in the human specimens from the TCGA PanCancer cohort, and found YB1 expression levels to be significantly (*p* < 0.0001) higher in AA patients (*n* = 182) when compared to CA patients (*n* = 749) (Figure 2C). Since mRNA levels may not always correlate with protein levels, we interrogated the relationship between YB1 mRNA and protein levels in the TCGA PanCancer cohort and found a significant positive correlation (*p* = 1.77 × 10^−15^, Spearman) between mRNA and protein levels (Appendix A). We also found YB1 protein expression levels to be significantly (*p* = 2.03 × 10^−3^) higher in AA BC tumors when compared to Cas (Appendix A). We further confirmed these findings in a separate TNBC cohort that we acquired from the MetroHealth tumor biobank which contains clinical and survival information on 45 TNBC patients (25 AA and 20 CA). RT-PCR analyses showed that YB1 expression levels were significantly (*p* = 0.002) higher in AA TNBC tumors when compared to their CA counterparts (Figure 2D). Thus, we show that YB1 expression levels are associated with the AA race.

### 3.3. Loss of YB1 Inhibits the Oncogenic Behavior of AA TNBC Cell Lines In Vitro

To better understand how the oncogenic activity of YB1 contributes to TNBC disparities, we sought to determine the effect of inhibition of YB1 expression on the behavior of TNBC cell lines of both AA and CA origins. We applied the Synthego CRISPR-based knockout strategy that is based on a ribonucleoprotein (RNP) mix which contains both sgRNAs oligos and the translated Cas9 protein, and is used to deliver the sgRNAs to the nucleus of target cells via electroporation without the need for virus infection and drug selection. We used a pool of three different sgRNAs to target YB1 in three AA and three CA TNBC cell lines. Western blot analyses showed a KO efficiency of more than 90% in all six cell lines (Figure 3A and Appendix A). The KO efficiency was sustained after more than 15 passages in all six lines supporting the effectiveness the KO strategy. In some cases, after serial passaging of the KO cells, we observed progressive re-expression of YB1, and a simple second electroporation with the sgRNA RNP mix was sufficient to eliminate the residual YB1 expression.

The AKT pathway is widely documented to be affected by the loss of YB1 expression [60], which we confirmed in our YB1-KO cells where phosphorylation levels of AKT were decreased in all six cell lines as a result of the loss of expression of YB1 (Figure 3A). Next, we used the colony formation assay as an in vitro indicator assay for the oncogenicity of cancer cells, and found the loss of YB1 expression to significantly inhibit the colony formation potential of all six cell lines (Figure 3B,C). While colony formation was reduced in all six TNBC cell lines, the fold reduction in the AA cell lines ranged from 1.8 to 3 and averaged a 2.4-fold inhibition when compared to the CA cell lines, where the fold reduction ranged from 2.5 to 3.25 with and an average of 2.95-fold inhibition (Figure 3C). These data indicate that the CA TNBC cells are more sensitive to the loss of YB1 activity when compared to their AA counterparts, which show more addiction to YB1.

### 3.4. Loss of YB1 Inhibits Tumor Growth and Metastasis of AA TNBC Cell Lines In Vivo

To further assess the effects of loss of YB1 expression on tumor growth and metastasis of TNBC cells in vivo, mammary fat pads of female Balb/C mice were inoculated with parental syngeneic 4T1 TNBC cells or their YB1-KO derivatives, and tumor growth was assessed over 5 weeks. Loss of YB1 significantly inhibited the growth of primary tumors (Figure 4A,B). While every mouse implanted with the parental 4T1 or the YB1-KO cells developed tumors (100% incidence), tumor latency was more than 1 week longer in the mice implanted with the YB1-KO 4T1 cells (Figure 4A). Tumor burden, as assessed by tumor volume (Figure 4B), was significantly lower (*p*  <  0.0001) in mice implanted with the YB1-deficient cells compared to their parental counterparts. Therefore, we show that loss of YB1 expression inhibits tumor growth, in vivo. Proliferation assays in 2D cultures showed no significant difference in cell growth between parental and YB1-KO (Appendix A). Therefore, the differences in tumor growth were rather the result of inhibition of YB1 oncogenic activity. We also used Western Blot analyses of tumor lysates to confirm the sustained loss of YB1 expression in the tumors derived from the YB1-KO cells as these tumors grew in a slower rate as compared to the tumors derived from the parental cells (Appendix A). 

Next, we sought to investigate whether inhibition of YB1 expression influences BC metastasis to the lungs. In the spontaneous metastasis assay, after the mice were sacrificed and the lungs were removed, and metastasis foci were counted. The number of lung metastasis foci was more than five-fold (*p*  <  0.001) lower in mice injected with the YB1-deficient 4T1 cells when compared to their parental counterparts (Figure 4C). These results show that YB1 plays a critical role in the growth, invasion, and metastasis of TNBC tumors, as demonstrated here in the tumor growth and the spontaneous metastasis assays.

Next, we assessed whether loss of YB1 would differentially affect tumor growth and metastasis in human AA vs CA TNBC cell lines. We used one CA TNBC cell line (MDA-MB-231) and one AA TNBC cell line (MDA-MB-468) and the immuno-compromised mouse BC model NOD-scid-IL2Rgamma knockout (NSG) mice. While mice implanted by either the parental MDA-MB-231 or MDA-MB-468 developed tumors (Figure 4D), tumor burden was significantly higher in the mice implanted with the AA MDA-MB-468 TNBC cells (Figure 4D, black graph) when compared to the AA MDA-MB-231 cells (Figure 4D, grey graph), which suggests the more aggressive nature of TNBC tumors of AA origin. Additionally, the loss of YB1 expression inhibited the tumor growth of both MDA-MB-231-YB1-KO cells (Figure 4D, yellow graph) and MDA-MB-468-YB1-KO cells (Figure 4D, red graph). Tumor growth inhibition was, however, more pronounced in the CA MDA-MB-231 cells that are more sensitive to loss of YB1 when compared to AA MDA-MB-468 cells that are more resistant to loss of YB1 (Figure 4D, compare red graph to yellow graph). Thus, our data suggest a difference in sensitivity to loss of YB1 in tumor growth between AA and CA TNBC cells in vivo. These findings were also confirmed with respect to spontaneous lung metastasis (Figure 4E,F), where lung metastasis burden was ~2-fold higher with the AA MDA-MB-468 cells when compared to the CA MDA-MB-231 cells (Compare black bar of Figure 4E to grey bar of Figure 4F). Finally, while loss of YB1 significantly inhibited the metastasis of both AA MDA-MB-468 cells (Figure 4E, red bar) and CA MDA-MB-231 cells (Figure 4F, yellow bar), lung metastasis was reduced by ~3-fold in the AA MDA-MB-468 cells (Figure 4E) when compared to more than 8-fold in the sensitive CA MDA-MB-231 cells (Figure 4F). Together, these data, strongly suggest a critical role of YB1 in the aggressiveness of AA TNBC in vitro and in vivo.

### 3.5. YB1 Is Associated with the Cancer Stem Cell Phenotype in AA TNBC

In our previously published studies [31,49] we showed that YB1 is required for the regulation of cancer stem cell phenotypes by modulating the expression levels of cancer stem cell (CSC) transcription machinery, including Oct4, Sox2, and Nanog. To determine whether these CSC transcription factors are differentially expressed between AA and CA TNBC tumors and cell lines, we interrogated the TCGA PanCancer cohort and found Oct4 (Figure 5A), Nanog (Figure 5B), and Sox2 (Figure 5C) to be significantly (*p* < 0.001) highly expressed in tumors of the basal (TNBC) breast cancer subtype when compared to tumors of Her2, Luminal-A, or Luminal-B subtypes. Oct4 (Figure 5D), Nanog (Figure 5E), and Sox2 (Figure 5F) were also significantly disparately highly expressed in AA tumors when compared to their CA counterparts. We have also shown that the WAVE3-YB1 signaling axis is required for the activation of the CSC phenotype in TNBC [31,49]. Here, WAVE3 is also significantly (*p* < 0.001) highly expressed in the tumors of the basal subtype (Figure 5G) and is significantly (*p* < 0.001) highly expressed in AA than in CA tumors (Figure 5H). Thus, we show that not only YB1 is highly expressed in AA versus CA TNBC tumors, but YB1-associated CSC genes are also disparately associated with TNBC tumors of AA origin, which may in part explain the aggressivity of these tumors in the AA population. We confirmed these findings in our MetroHealth TNBC cohort, where YB1 (Figure 2E), Oct4 (Figure 5I), Nanog (Figure 5J), Sox2 (Figure 5K), and WAVE3 (Figure 5L) are significantly highly expressed in AA tumors when compared to their CA counterparts.

### 3.6. YB1 Is Associated with Chemoresistance in AA TNBC Both In Vitro and In Vivo

YB1 was shown to be associated with chemotherapy resistance in several types of cancer, including BC (reviewed in [16]). We interrogated the Cancer Cell Line Encyclopedia (CCLE) for the IC50 of three of the most commonly used chemotherapeutic agents for the treatment of patients with TNBC (Cisplatin, Docetaxel and Doxorubicin) in cell lines of the TNBC subtype; these included three AA and three CA TNBC cell lines. We found the IC50 values, which are indicative of level of resistance to a given drug, to be consistently higher in AA cell lines when compared to their CA counterparts for cisplatin (Figure 6A,B), doxorubicin (Figure 6C,D) and docetaxel (Figure 6E,F). Since YB1 expression levels were also significantly elevated in AA cell lines when compared to their CA counterparts (Figure 2E), these results show a clear positive correlation between YB1 expression and resistance to chemotherapies. Gene set enrichment analysis (GSEA) of the TCGA breast cancer cohort confirmed our findings, where we found a significant (*p* = 0.003) enrichment of genes that are associated with docetaxel resistance in BC tumors that have increased levels of YB1 expression (Figure 6G) [61]. To investigate the role of YB1 in the chemoresistance in TNBC, we treated MDA-MB-468 with doxorubicin for 48 h. and found that this transient treatment resulted in the inhibition of YB1 expression and its phosphorylation levels (Figure 6H and Appendix A). However, in MDA-MB-468 that were generated to be permanently resistant to doxorubicin (Dox-Res), we found this chronic exposure to doxorubicin to instead activate both the expression and phosphorylated levels of YB1 (Figure 6H and Appendix A). These finding were confirmed in MDA-MB-468 that were generated to be resistant to cisplatin (Cis-Res) and found both the expression and phosphorylation of YB1 to be increased in the Cis-Res cells when compared to their control counterparts (Figure 6I and Appendix A). Thus, we show that chronic exposure to chemotherapy induces YB1 expression and phosphorylation. Next, we injected either the parental or the cisplatin-resistant (Cis-Res) MDA-MB-468 cells into the mammary fat pads of female NSG mice. At 21 days post implantation, mice from both the groups received two doses of cisplatin (1 mg/kg, IP) per week for 3 weeks, and tumor growth was followed for several weeks (Figure 6J). We found that treatment with cisplatin resulted in inhibition of tumor growth in both groups for the first three weeks following the treatment; after this three-week period, the mice that were injected with the Cis-res cells showed a tumor growth that was much faster than the mice injected with the parental control cells. After 70 days of follow-up, the average volume of the tumors derived from the parental cells did not exceed 150 mm^3^, while that of the tumors derived from the cisplatin-resistant cells rapidly reached the maximal tumor burden of more than 1000 mm^3^ (Figure 6J). Therefore, we speculate that the increased YB1 expression in the Cis-resistant cells, which is induced by chronic exposure to chemotherapy, confers these cells with an enhanced tumor growth potential and makes them refractory to chemotherapy treatment. Together, our data suggest a novel positive feedback loop between YB1 expression and chemoresistance, therefore creating a vicious oncogenic environment which activates a sustained tumor growth and metastasis, which is more active in AA TNBC tumors, and ultimately contributes to the disparities in health outcomes in AA women with TNBC disease. 

## 4. Discussion

Breast cancer is the second leading cause of cancer-related death in both Caucasian American (CA) and African American (AA) women. The most devastating subtype of BC is TNBC. TNBC accounts for about 15–20% of all invasive breast cancer cases in the United States [62]. By definition, TNBCs lack the expression of estrogen and progesterone hormone receptors and presents with low copies of Her2 (EGFR2). Whereas targeted therapies are available for other breast cancer subtypes, including hormone receptor-positive and HER-2 amplified breast cancers, there aren’t any FDA-approved targeted therapies for TNBC, and treatment options are limited to chemotherapy that is usually ineffective especially in later stages. There have not been any meaningful additions to our TNBC armamentarium for a while, and the recent excitement of a breakthrough, with the accelerated approval of the immune checkpoint inhibitor atezolizumab, was short-lived when the manufacturer voluntarily withdrew the approval. The cancer mortality disparity between AA and CA women is alarming, with death rates being 40% higher in AAs (28.4 vs. 20.3 deaths per 100,000), and in AAs aged less than 50 years, it is tragic with the death rate that is double that of CA and is the highest among all race groups [63]. Reasons for these disparities include AA women presenting at a younger age, where 33% are diagnosed aged less than 50 years when compared to 21.9% of CA women [64]. AA women are also more likely to delay breast cancer screening mammography, as well as to present with an advanced stage, face an average waiting time to the initiation of treatment that is double that of CA women face, and less likely to have surgery [65]. 

While environmental factors and socioeconomic status play undeniable roles in driving poor outcomes in AAs, a key gap in knowledge remains regarding the biological and molecular mechanisms that interact with these extrinsic factors to impact treatment response and clinical outcomes. Studying disparities in cancer-related morbidity and mortality between different racial and ethnic groups is complex and multifaceted. Examining the potential relationships between social and biological factors is of paramount importance. From the time of the inception of the Surveillance, Epidemiology, and End Results (SEER) program in 1973 until 1983, breast cancer mortality in CA and AA women was almost the same; however, a divergence in mortality started following 1983, which coincided with the introduction of tamoxifen for hormone receptor-positive cancers [66]. A combination of these factors contributed to the fact that AA TNBC patients end up with significantly worse outcomes when compared to their CA counterparts. While environmental factors and socioeconomic status play undeniable roles in driving poor outcomes in AAs, a key gap in knowledge remains regarding the biological and molecular mechanisms that interact with these extrinsic factors to impact treatment response and clinical outcomes. In the present study, we used bioinformatics and biostatistics analyses to interrogate both public and local BC cohorts to show that YB1 may be a driver of BC health disparities in AA women. We found that not only YB1, but its associated cancer stem cell and invasion-metastasis gene signatures, including WAVE3, are also associated with worst disease outcomes in AA patients with TNBC tumors.

We supported our findings with a combination of in vitro genetic manipulations of YB1 (CRISPR/Cas-mediated knockout) and treatment with chemotherapeutic agents, as well as in vivo mouse models of TNBC to show that the loss of YB1 expression inhibits its oncogenic activity both in vitro (colony formation) and in vivo (inhibition of tumor growth and metastasis) in TNBC cell lines of both CA and AA origin. However, inhibition of YB1 oncogenic activity was more pronounced in cell lines of CA origin when compared to their AA counterparts. This suggested to us that YB1 might be more active in AA TNBC cell lines than in their CA counterparts, since the residual YB1 expression in the AA cell lines after CRISPR-KO may be sufficient to continue driving YB1 oncogenic activity in AA cell lines. This suggestion is supported by the fact that YB1 is also involved in several oncogenic activities, including in the regulation of the CSC phenotype [31,45,49,61,67]. In fact, our data show that expression levels of CSC markers Nanog, Sox2, and Oct4, as well as their upstream regulator, WAVE3, are significantly higher in AA TNBC tumors when compared to CAs (Figure 5). Our published studies [31] also showed that loss of either YB1 or WAVE3 expression inhibits the expression of these CSC markers, albeit not completely. Therefore, we posit that the residual levels of YB1 and its associated CSC markers is responsible for the enhanced aggressivity of the AA TNBC cell lines both in vitro (Figure 3) and in vivo (Figure 4). YB1 is also widely documented as a major player in the regulation of chemoresistance in several types of cancer, including BC (reviewed in [16]), which reenforces the association between YB1 and chemoresistance in AA TNBC tumors, given that AA TNBC are more prone to develop resistance to standards of care chemotherapies than their CA counterparts (reviewed in [16]). Our data show a clear and significant positive correlation between YB1 expression and the IC50 values of the chemotherapeutic drugs that are commonly used to treat TNBC tumors, in TNBC cell lines of AA origin (Figure 6). We further identify a positive aggressive feedback loop between YB1 expression and resistance to chemotherapy to drive the tumor growth of AA tumors. While short-term treatment with chemotherapy seems to inhibit both YB1 expression and its phosphorylation levels, chronic exposure to these therapeutic drugs results in the reactivation of YB1 expression and its oncogenic activities: Cisplatin-and doxorubicin-resistant MDA-MB-468 cells express more YB1 and phopho-YB1 when compared to their parental cells, and tumors derived from the cisplatin-resistant AA MDA-MB-468 TNBC cells grow faster and bigger than those derived from their parental counterparts, even after chemotherapy treatment (Figure 6). These data strongly support the interplay between YB1 and chemoresistance that drives the aggressivity of AA TNBC tumors.

## 5. Conclusions

Taken together, our study supports the role of YB1 as a major molecular contributor to the health outcome disparities observed in AA women with TNBC tumors. A simple PubMed search with the keywords YB1 and cancer resulted in more than 600 publications, which demonstrate the significant interest in investigating YB1 in the context of cancer biology. However, none of these studies attempted to investigate the potential role of YB1 in health disparities. Our study is the first to demonstrate the potential contribution of YB1 to TNBC health disparities in AA women. In fact, the interrogation of public databases of transcriptomic TCGA datasets revealed that YB1 is associated with health disparities in AA women not only in TNBC tumors, but in several other cancers of epithelial origin, such as colon and ovarian cancers. This clearly demonstrates a global association of YB1 with AA health disparities and justifies the urgent need for the development of targeted therapies against YB1 that would benefit not only the general population but the AA patient population, which is disparately affected by the cancer burden.

## Figures and Tables

**Figure 1 cancers-13-06262-f001:**
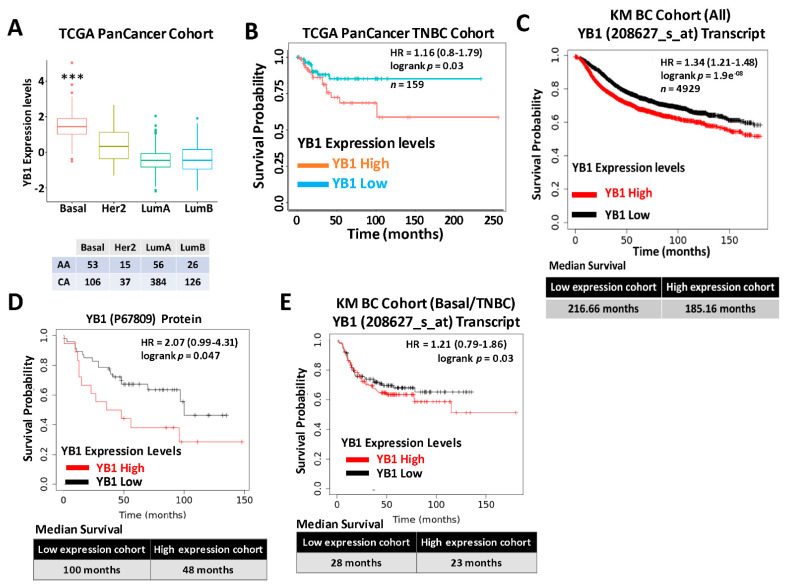
(**A**) Quantification of Yb1 mRNA expression levels based on log2 RSEM values batch normalized from Illumina HiSeq RNASeq data by breast cancer subtype in the breast cancer BRCA patient data from TCGA PanCancer Atlas. Expression levels of YB1 are significantly higher (*** *p* < 0.0001, Wilcoxon) in the basal (TNBC) subtype when compared to other BC subtypes. (**B**) Survival analysis based on YB1 expression levels (median values) in the basal subtype of the TCGA BRCA PanCancer Atlas cohort. YB1 activation in TNBCs is associated with poorer overall survival (*p* = 0.049). (**C**) KM plot correlating survival of 4929 BC patients with YB1 mRNA expression levels. High YB1 expression levels correlate with poor survival probability in BC patients (*p* < 1.9 × 10^−8^). (**D**) KM plot correlating the survival of BC patients with YB1 protein expression levels. High YB1 expression levels correlate with poor survival probability in BC patients (*p* = 0.047). (**E**) KM plot correlating the survival of TNBC patients with YB1 mRNA expression levels. High YB1 expression levels correlate with poor survival probability in TNBC patients (*p* = 0.03).

**Figure 2 cancers-13-06262-f002:**
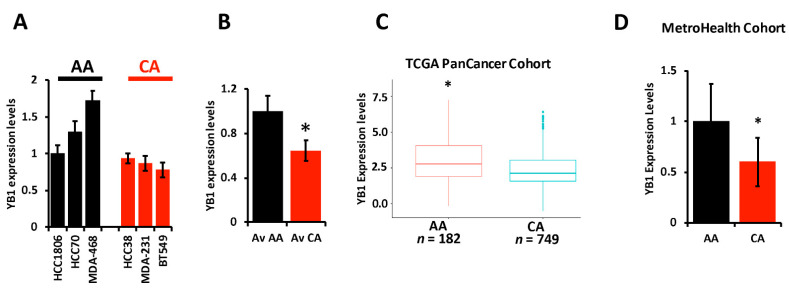
(**A**) YB1 mRNA transcript quantification in AA and CA TNBC cell lines was determined by qt-RT-PCR and data was plotted as the fold change in HCC1806. (**B**) Average YB1 mRNA transcripts in AA and CA TNBC cell lines was plotted as the fold change in AA cell lines. YB1 transcript expression levels are significantly higher (* *p* < 0.001, Student ’*t* test) in AA TNBC cell lines when compared to their CA counterparts. (**C**) Comparison of YB1 expression levels, levels based on log2 RSEM values batch normalized from Illumina HiSeq RNASeq data, between AA and CA BC patients from the TCGA BRCA PanCancer Atlas cohort. YB1 expression levels are significantly higher (* *p* < 0.01, Wilcoxon) in AA when compared to CA BC tumors. (**D**) Comparison of YB1 expression levels between CA and AA TNBC from the MetroHealth TNBC cohort, as determined by qt-RT-PCR. Data was plotted as the fold change to AA tumors. YB1 expression levels are significantly higher (* *p* < 0.01, Student ’*t* test) in AA when compared to CA BC tumors.

**Figure 3 cancers-13-06262-f003:**
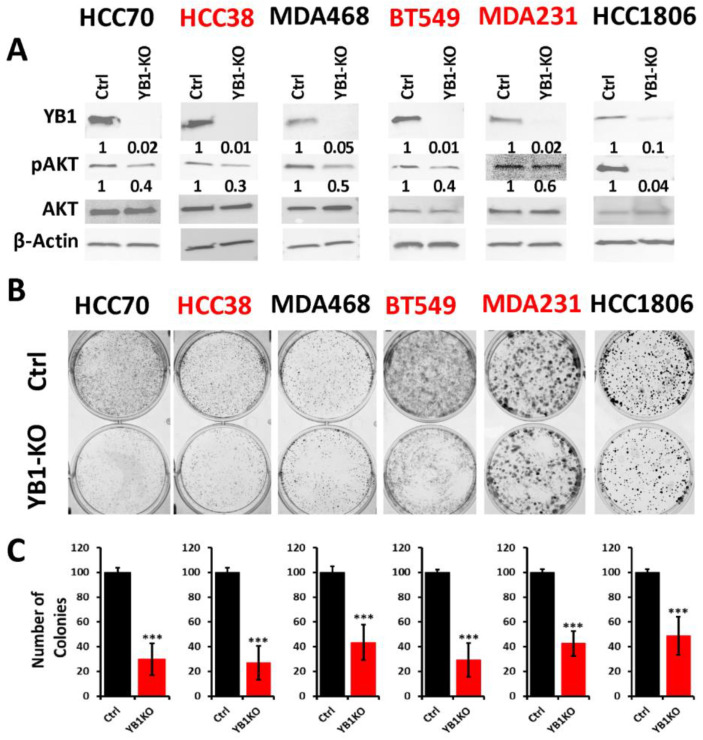
(**A**) Western Blot analysis of protein lysates from parental (Ctrl) AA (black font) and CA (red font) TNBC cell lines and their YB1-KO derivatives, that were probed with the indicated antibodies. β-Actin was used a loading control. (**B**,**C**) Colony formation assay. The numbers under the bands represent the fold change of the Western Blot signal normalized to that of the parental cell line, as determined by densitometry analyses from one representative of 3 replicate blots. (**B**) Representative images of colony formation of parental and YB1-KO AA (black font) and CA (red font) TNBC cell lines. (**C**) Quantification of the number of colonies in each plate. Data is plotted as the percentage change from the control cells. Data shown are representative of 3 replicates (***, *p* < 0.0001, Student ’*t* test).

**Figure 4 cancers-13-06262-f004:**
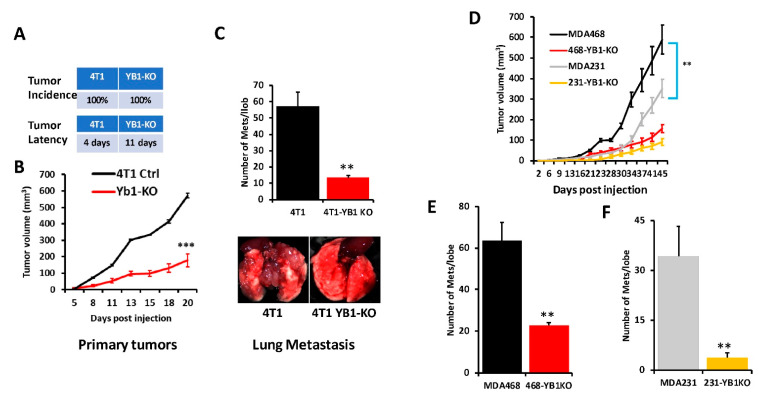
(**A**) Incidence (top) and latency (bottom) of tumor growth of prenatal 4T1 or its YB1-KO derivative cells that were inoculated in the mammary fat pads of female Balb/C mice (*n* = 5 mice per group). (**B**) Quantification of tumor volume of tumors generated from the inoculation of control 4T1 cells or their YB1-KO derivatives into the mammary fat pads of female Balb/C mice. (**C**, **top panel**) Quantification of lung metastasis nodules from the animal experiment described in (**B**). The bottom panel shows representative lungs from each mouse group. (**D**) Quantification of the tumor volume of tumors generated from the inoculation of control MDA-MB-231 (grey graph), MDA-MB-468 (black graph) cells or their respective YB1-KO derivatives (yellow and red graphs, respectively) into the mammary fat pads of female NSG mice (n = 5 mice per group). (**E**,**F**) Quantification of lung metastasis nodules from the MDA-MB-468 (**E**) and the MDA-MB-231 animal experiment described in (**D**) (**, *p* < 0.001, Student ’*t*-test).

**Figure 5 cancers-13-06262-f005:**
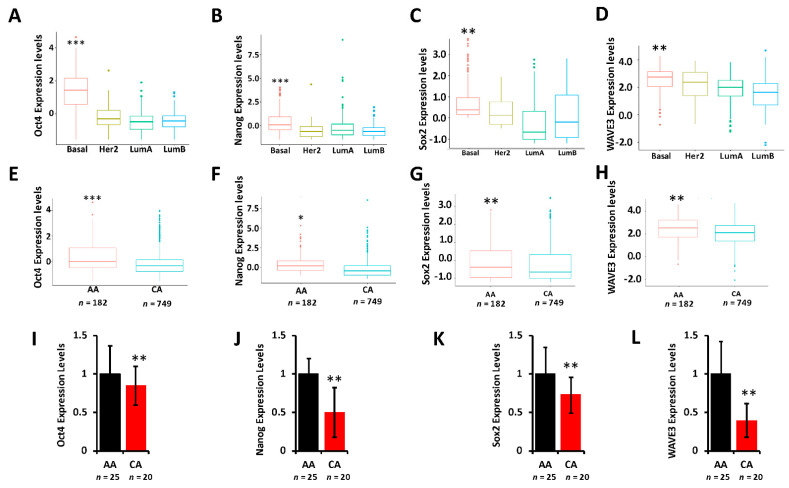
(**A**–**D**) Comparison of the expression levels of Oct4 (**A**), Nanog (**B**), Sox2 (**C**), and WAVE3 (**D**) between the BC subtypes in the TCGA BRCA PanCancer Atlas cohort. (**, *p* < 0.001; ***, *p* < 0.0001, Wilcoxon). (**E**–**F**) Comparison of the expression levels of Oct4 (**E**), Nanog (**F**), Sox2 (**G**), and WAVE3 (**H**) between AA and CA BC patients in the TCGA BRCA PanCancer Atlas cohort. (*, *p* < 0.01; **, *p* < 0.001; ***, *p* < 0.0001, Wilcoxon). In all cases, mRNA expression levels were based on log2 RSEM values batch normalized from Illumina HiSeq RNASeq data. (**I**–**L**) Comparison of the expression levels of Oct4 (**I**), Nanog (**J**), Sox2 (**K**), and WAVE3 **(L**) between AA and CA BC patients in the MetroHealth cohort, as determined by qt-RT-PCR. Data were plotted as fold change to AA tumors. (* *p* < 0.01, **, *p* < 0.001; ***, *p* < 0.0001, Wilcoxon).

**Figure 6 cancers-13-06262-f006:**
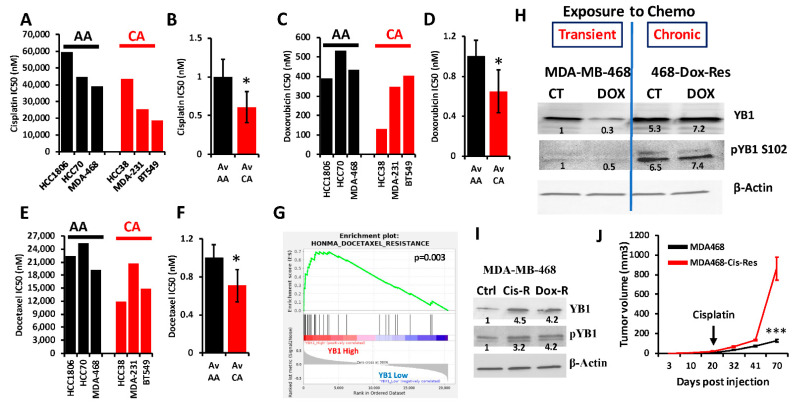
(**A**–**E**) Comparisons of the IC50 values of cisplatin (**A**,**B**), doxorubicin (**C**,**D**) and docetaxel (**E**,**F**) between AA (black bars) and CA (red bars) TNBC cell lines. The IC50 values were derived from the CCLE database. Average IC50 in the AA and CA TNBC cell lines was plotted as fold change to AA cell lines for cisplatin (**B**), doxorubicin (**D**) and docetaxel (**F**), (*, *p* < 0.01, Fisher’s Exact). (**G**) Gene set enrichment analysis of the TCGA BRCA PanCancer Atlas dataset shows that the gene signature associated with docetaxel resistance is enriched in the BC tumors that express high levels of YB1. (**H**) Western Blots with the indicated antibodies of protein lysates from parental MDA-MB-MB-468 cells or those that were treated with doxorubicin for 48 h (transient treatment), or MDA-MB-468 cells that were generated to be permanently resistant (chronic) to doxorubicin (468-Dox-res). The numbers under the bands represent the fold change of the Western Blot signal normalized to that of the parental cell line. (**I**) Western Blots with the indicated antibodies of protein lysates from parental MDA-MB-468, their cisplatin resistant (Cis-R) or doxorubicin (Dox-R) derivatives. β-Actin was used as a loading control. The numbers under the bands represent the fold change of the Western Blot signal normalized to that of the parental cell line. (**J**) Quantification of the tumor volume of tumors generated from the inoculation of parental MDA-MB-468 cells (black graph), or their cisplatin resistant (Cis-R) derivatives into the mammary fat pads of female NSG mice (n = 5 mice per group). Treatment with cisplatin started 21 days post inoculation. (***, *p* < 0.0001, Student ’*t* test).

**Table 1 cancers-13-06262-t001:** Breast cancer patients’ tumors breakdown by race and BC subtype from the TCGA PanCancer Atlas database.

Race	Basal	Her2	LumA	LumB	NA	Normal Like	Grand Total
AfAm	53	15	56	26	23	9	182
Asian	7	15	20	16	1	1	60
NA	5	10	39	29	6	1	90
Native American		1					1
White	106	37	384	126	71	25	749
**Grand Total**	**171**	**78**	**499**	**197**	**101**	**36**	**1082**

## Data Availability

The data presented in this study are available on request from the corresponding author.

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
