# Peer review of "YB1 Is a Major Contributor to Health Disparities in Triple Negative Breast Cancer"

_cancers, 2021, doi:10.3390/cancers13246262_

Round 1
Reviewer 1 Report
The study by Rana PS et al. addressed a key question: what may be a possible contributing factor to a higher incidence rate of triple-negative breast cancer (TNBC) and higher breast cancer (BC) –related death rate in non-Hispanic black women across all race/ethnicity. From the biological standpoint, the authors took advantage of bioinformatics tools to identify the Y-box binding protein-1 (YB1) as a major factor responsible for TNBC disparities between African American and Caucasian American women. The authors further testified their hypothesis using genetic and pharmacological manipulation of YB1 in combination with in vitro cell lines and in vivo mouse models and found that YB1 enhanced tumorigenesis and metastasis through the activation of cancer stem cell biomarkers and promoted resistance to chemotherapeutic treatment. Overall, the manuscript is highly relevant to the field and the study itself was performed thoroughly and was described well although the findings are not surprising [BMC Cancer (2017) 17:201; Cancer Lett (2019) 452:119; Cancers (2020) 12:2795; J Lab Physicians (2018) 10:420].
Major comment:
This study basically relied on the YB1 transcript expression level to support its conclusion. It has been widely assumed that transcript level correlates with protein level. However, accumulating evidence suggests it may not be the case due to the complexity of gene expression regulation [Cell (2016) 165:535]. It will strengthen the quality of this manuscript by performing immunohistochemistry on those 45 human TNBC FFPE sections.
Minor comments:
- It is not clear how the authors defined low and high expression levels of YB1.
- It may be more relevant to present only the TNBC subset instead of the whole KM BC cohort in Fig 1C.
- Missing Fig1D legend.
- In Fig2A, it looks like that there is no difference inYB1 mRNA transcript levels between HCC1806 and HCC38. It is not clear how the authors performed the statistical analysis. The similar issue with Figs 6A, 6B & 6C .
- Typo “trough” in line 30. It should be “through” instead.
Author Response
Subject: Revision and resubmission of manuscript number: and response to the reviewers
Our Response
Dear Reviewer,
We would like to thank you for extending us the opportunity to submit a revised draft “YB1 is a major contributor to health disparities in triple negative breast cancer” for publication in Cancers Journal. We appreciate your time and efforts to review the article and provide constructive feedback. We have now addressed most if not all the issues raised by this Reviewer. We have performed new experiments and analyses to strengthen the study. These changes are marked in red text within the manuscript and are described in the point-by-point response to the Reviewer below.
Reviewer 1:
Comments and Suggestions for Authors
Comments for Rana et al
Overall, the manuscript is highly relevant to the field and the study itself was performed thoroughly and was described well.
Major Comment:
This study basically relied on the YB1 transcript expression level to support its conclusion. It has been widely assumed that transcript level correlates with protein level. However, accumulating evidence suggests it may not be the case due to the complexity of gene expression regulation [Cell (2016) 165:535]. It will strengthen the quality of this manuscript by performing immunohistochemistry on those 45 human TNBC FFPE sections.
Our Response: We agree with the Reviewer that in several instances the levels of the mRNA transcript do not always correlate with protein expression levels. Performing immunohistochemistry on our human TNBC FFPE sections will certainly provide a conclusive answer to the question regarding YB1. This effort will however be very time consuming since we have to start the process all over again and request fresh sections of these tumors from the pathology department, which will take weeks if not month to complete. As a more attractive and reliable alternative, we went back and analyzed the TCGA PanCancer data set that mRNA and protein information on more than 1000 BC tumors as compared to our cohort of 45 tumors. We found found a very significant positive correlation between YB1 mRNA transcript levels and YB1 protein expression levels: Tumors that express high levels of YB1 mRNA also express high levels of YB1 protein. This new data is now presented in the new Supplemental Figure 1 (Fig. S1) and described in the revised manuscript text in L307-309. More importantly, we also found that YB1 protein expression and phosphorylation levels are significantly higher in AA as compared to CA breast cancer women. This new data is now presented in the Supplemental Figure 2 (Fig. S2) and described in the revised manuscript text in L307-309. Finally, we performed survival probability analyses based on YB1 expression levels and found high YB1 expression levels to correlate with almost 50% decreased survival of TNBC patients. This new data is now presented in panel D of revised Figure 1, and described in in the revised manuscript in L273-274. With the new information, are now confident that these new data strengthened our overall conclusions and alleviate the concerns of this Reviewer.
Minor Comments:
- It is not clear how the authors defined low and high expression levels of YB1.
Our Response: For all the survival analyses, the median values were used as the defining point between high and low expression levels.
- It may be more relevant to present only the TNBC subset instead of the whole KM BC cohort in Fig 1C.
Our Response: This is a very valid point and we completely agree with the Reviewer. We have now performed survival analyses based on YB1 expression levels in only the TNBC subset instead of the whole KM BC cohort, and confirmed the finding that high YB1 expression levels correlate with decreased survival probability in TNBC patients. This new data is now presented in the new panel E of revised Figure 1, and described in in the revised manuscript in L273-274.
- Missing Fig1D legend.
Our Response: The revised manuscript has now gone through a thorough editing to make sure all the material is adequately described.
- In Fig2A, it looks like that there is no difference inYB1 mRNA transcript levels between HCC1806 and HCC38. It is not clear how the authors performed the statistical analysis. The similar issue with Figs 6A, 6B & 6C.
Our Response: We have now revised both Figure 2 and Figure 6 to address this concern. In revised Figure 2, we have included a new panel (Fig. 2B), where YB1 expression levels are compared between AA and CA TNBC cell lines by averaging YB1 expression levels in AA and CA cell lines. The difference is still very significant, where average YB1 in AA is significantly higher than that of CA cell lines.
The same process was applied to the IC50 of chemotherapeutic drugs in Figure 6.
- Typo “trough” in line 30. It should be “through” instead.
Our Response: The entire manuscript has now gone through a thorough proof editing.

Reviewer 2 Report
The manuscript by Rana et al. describes the role of YB1 protein as a major contributor to the health disparities observed in African American (AA) women having triple negative breast cancer (TNBC). Based on previous published data of the lab, the present work brings clear evidence of the importance of YB1 in the aggressiveness of TNBC. The experimental design is sound, the manuscript is coherent and well written; however, several concerns remain to be addressed.
Major comments:
- Some parts of the study are very exaggerative or speculative:
- Abstract and l. 502: “pharmacologic manipulation of YB1” suggests that specific inhibitors have been used, while in fact chemotherapy was observed to reduce YB1 expression and activity. Please correct.
- L. 361-383: By comparing two cell lines (one of AA and the other of CA origin) one cannot draw the conclusion that the observed difference “supports the more aggressive nature of TNBC tumors of AA origin”. This is highly speculative. Similarly, the statement “Thus, we show a difference in sensitivity to loss of YB1 in tumor growth between AA and CA TNBC cells in vivo” is a huge overstatement. Please re-interpret this paragraph.
- L. 424-447: The authors show that expression and phosphorylation of YB1 is increased in cisplatin-resistant MDA-MB-468 cells. They also show that in mice injected with cisplatin-resistant MDA-MB-468 cells cisplatin treatment does not inhibit tumor growth after three weeks. The conclusion that “our data identify a novel positive feedback loop between YB1 expression and chemoresistance” is not supported by these data because no correlation is shown. This part must be changed.
- Number of independent experiments and significance values are missing in several cases. E.g. western blots, Fig. 3 B.
- Please show proliferation data for each control and KO cell line. These are very important data and should be included in the paper (l. 339-342).
- The authors show “a difference in sensitivity to loss of YB1 in tumor growth between AA and CA TNBC cells in vivo” (l. 373-374). Is YB1 re-expressed in vivo along cell division?
Minor comments:
- Fig. 2 A and Fig. 6 A-C: Please show which bars were compared. If an average of the three AA and three CA values were compared, please complete the figures with bars showing the average and place significance markers accordingly.
- Please use the same scale (or %) on Y-axes in charts of Fig. 3 A.
- What are the units in the Y-axes in Fig. 5?
Author Response
Subject: Revision and resubmission of manuscript number: and response to the reviewers
Our Response
Dear Reviewer,
We would like to thank you for extending us the opportunity to submit a revised draft “YB1 is a major contributor to health disparities in triple negative breast cancer” for publication in Cancers Journal. We appreciate your time and efforts to review the article and provide constructive feedback. We have now addressed most if not all the issues raised by this Reviewer. We have performed new experiments and analyses to strengthen the study. These changes are marked in red text within the manuscript and are described in the point-by-point response to the Reviewer below.
Reviewer 1:
Comments and Suggestions for Authors
Comments for Rana et al
Based on previous published data of the lab, the present work brings clear evidence of the importance of YB1 in the aggressiveness of TNBC. The experimental design is sound, the manuscript is coherent and well written; however, several concerns remain to be addressed.
Major Comments:
- Some parts of the study are very exaggerative or speculative:
- Abstract and l. 502: We have now revised this statement to reflect the sgRNA-KO and chemotherapy treatments were used manipulate YB1 expression and activity.
- L. 361-383: We have now revised these sentences to reflect a more suggestive rather than definitive conclusions.
- L. 424-447: We have also revised this sentence to state: Thus, we show that chronic exposure to chemotherapy induces YB1 expression and phosphorylation.
- Number of independent experiments and significance values are missing in several cases. E.g. western blots, Fig. 3 B. We have now revised the figure legends to clearly state that the presented data are representative of at least three replicate experiments and the values shown are averages of these three independent experiments.
- Please show proliferation data for each control and KO cell line. These are very important data and should be included in the paper (l. 339-342).
The proliferation data of all 6 TNBC cell lines and their YB1-KO derivatives are now presented in if the new Supplemental Figure 3 (Fig. S3), and described in the revised manuscript in L367-369.
- The authors show “a difference in sensitivity to loss of YB1 in tumor growth between AA and CA TNBC cells in vivo” (l. 373-374). Is YB1 re-expressed in vivo along cell division?
We appreciate this valid comment form the Reviewer. We have now performed WB analyses of protein lysates of tumors derived from the control and the YB1-KO cells from MDA-MB-231, MDA-MM-468 and 4T1 cells. We found that YB-1-KO was still sustained at similar levels in the tumors derived from both the AA and CA YB1-KO cells, further supporting our conclusions. These new data is now presented in the new Supplemental Figure 4 (Fig. S4), and described in the revised manuscript in L369-373.
Minor Comments:
- 2 A and Fig. 6 A-C: Please show which bars were compared. If an average of the three AA and three CA values were compared, please complete the figures with bars showing the average and place significance markers accordingly.
We have now revised both Figure 2 and Figure 6 to address this concern. In revised Figure 2, we have included a new panel (Fig. 2B), where YB1 expression levels are compared between AA and CA TNBC cell lines by averaging YB1 expression levels in AA and CA cell lines. The difference is still very significant, where average YB1 in AA is significantly higher than that of CA cell lines.
The same process was applied to the IC50 of chemotherapeutic drugs in Figure 6.
- Please use the same scale (or %) on Y-axes in charts of Fig. 3 A.
We believe the Reviewer meant charts of Fig. 3C. We have now revised these charts and the number of colonies are now presented as a % of the total colonies, where the control cells represent 100% and the YB1-KO are presented and % change of the controls.
- What are the units in the Y-axes in Fig. 5?
In all the analyses derived from the TCGA data sets, the units used for the expression levels are presented as the log2 of RSEM values batch normalized from Illumina HiSeq RNASeq data. RSEM is a commonly used unit for RNASeq expression. It is essentially a way to quantify gene expression based on the number of reads mapped to a particular transcript and is proportional to the abundance of a given transcript.

Round 2
Reviewer 1 Report
Thanks to the authors to address all my concerns and improve the manuscript quality. I have no further comments.
Reviewer 2 Report
The authors have adequately addressed my comments.